Language, Artificial Intelligence, and Digital Equity in North East India: A Quantitative and Policy Analysis of Low-Resource Language Exclusion, Educational Infrastructure, and Climate Knowledge System

Amme Shirisha, Baksheesh Sachar
shirishamadgul84@gmail.com, baksheesh22@gmail.com

**Abstract:**

North East India is home to over 220 tribal and indigenous languages, with a rich cultural heritage. Yet, it is systematically underrepresented in the data architectures that underpin artificial intelligence (AI), its languages are low-resource, its schools are digitally under-equipped, and its indigenous ecological knowledge exists almost entirely in oral, non-digitized forms. Drawing on quantitative analysis of UDISE+ 2024-25 school education data, Census 2011 language statistics, natural language processing (NLP) research from the WMT 2024 Low-Resource Indic Language Translation Shared Task, and policy analysis of the National Education Policy 2020, the Bhashini platform, and the IndiaAI Mission 2024, this paper presents six empirically grounded findings. Most importantly, the states where the dominant language is constitutionally scheduled average 19 percentage points higher internet access in schools than states where the dominant language is non-scheduled, a structural relationship between language recognition and digital access that has not been previously quantified at the sub-national level. The paper proposes a three-pillar integrated framework for Language AI, Education, and Climate Resilience and argues that investment in AI systems designed for community languages, rather than high-resource languages for local deployment, is an important factor for equitable digital development in the region.

**Keywords:** *Northeast India, low-resource languages, artificial intelligence, natural language processing, Bhashini, UDISE+, tribal education, indigenous ecological knowledge, climate resilience, digital equity*

1. **Introduction**

Despite the region's extraordinary diversity, its communities face a compounding structural disadvantage in the age of artificial intelligence. Most of its languages are classified as low-resource languages (LRLs), languages with insufficient digitized text, audio, or annotated data to train standard natural language processing (NLP) models (Nekoto et al., 2020). Only five languages spoken in Northeast India have full or partial coverage under India's flagship AI for language platform, Bhashini: Assamese, Manipuri, Bodo, Bengali, and Nepali, all of them constitutionally scheduled languages (PIB, 2025). The dominant languages of four states, Mizo (Mizoram), Khasi (Meghalaya), and the 16 Naga languages (Nagaland), and dozens of Arunachal languages have effectively zero AI infrastructure (WMT 2024; Bhaskar & Krishnamurthy, 2024).

This paper presents a systematic analysis of this exclusion in terms of the relationship between language recognition status and digital access, demonstrated through an original analysis of UDISE+ 2024-25 data correlated with Census 2011 language statistics. Then state-by-state digital infrastructure deficit in NE India's schools, quantified from UDISE+ 2024-25. Eventually looks at the relationship between indigenous ecological knowledge (IEK), preserved almost entirely in low-resource languages, and AI-enabled climate resilience, a connection illuminated by documented cases from Mizoram (Chinlampianga, 2011) and supported by global frameworks from UNESCO (2021), UNDP (2012), and IPCC (2022).

The central argument is that this AI deployment in Northeast India cannot be effective in any one domain without the other two. Language AI without an educational infrastructure has no delivery mechanism. Educational AI designed without a mother tongue excludes the communities it claims to serve. Climate AI without indigenous language knowledge misses the oldest and most ecologically precise dataset available. The three dimensions form a single integrated system, and the policy architecture required to address them, NEP 2020, Bhashini, IndiaAI Mission 2024, Samagra Shiksha, must be understood and evaluated as such.

## 2. Methodology

This study employs a qualitative secondary research design using narrative review and quantitative secondary data analysis. Primary quantitative data is drawn from the UDISE+ 2024-25 report published by the Department of School Education and Literacy, Ministry of Education, Government of India. Language coverage data is drawn from a cross-reference of Bhashini's publicly declared language portfolio against the language landscape documented in the Census 2011 Language Atlas, and is analyzed using the Mallikarjun (2020) multilingualism study. NLP corpus size data is drawn from the WMT 2024 Low-Resource Indic Language Translation Shared Task, with findings from IIIT Hyderabad (Bhaskar & Krishnamurthy, 2024) and IIT Madras (Sayed et al., 2024) providing quantified evidence of data scarcity.

Sources were selected for their thematic relevance to the intersection of language preservation, digital education, and climate resilience, with particular attention to low-resource language contexts, indigenous communities, and South and Southeast Asian settings comparable to those of Northeast India. The study acknowledges the limitations inherent in secondary research, including the sparse body of AI literature specific to NE India, and treats findings from comparable contexts as indicative rather than directly transferable.

## 3. Linguistic Diversity and Classification

Most of the North Eastern languages belong to four major families i,e, Tibeto-Burman (numerically dominant, spread across all seven sister states), Austro-Asiatic (Khasi, Munda group in Meghalaya), Indo-Aryan (Assamese, Bengali, Nepali), and Tai-Kadai (spoken in parts of Assam and Arunachal Pradesh) (Acharya, 2023). Census 2011 data reveal a striking variation in the demographic weight of these languages. In Mizoram, Lushai/Mizo is the mother tongue of 73.2% of the population (Mallikarjun, 2020). In Meghalaya, Khasi accounts for 46.6% of speakers and Garo for a further substantial share. In Nagaland, no single Naga language exceeds 15% of the population, with Konyak at 12.3% as the largest single group — reflecting the exceptional fragmentation of the Naga language cluster into 16 or more distinct languages (Mallikarjun, 2020). Population projections from the Indian Institute of Population Sciences estimate that by 2061, speaker communities for Bodo, Khasi, Garo, Lushai/Mizo, and Tripuri will each exceed 10 lakh speakers (IIPS, 2021), making early AI investment not merely equitable but strategically rational.

### *State by State Digital Infrastructure: Key Findings*

Table 1 presents the extracted UDISE+ 2024-25 data for all eight NE states across the five key digital infrastructure metrics, with percentage-point gap scores computed against the national average.

*Table 1: NE India School Digital Infrastructure vs. National Average, UDISE+ 2024-25*

| State | Schools | Internet % | Computer % | Smart Class % | ICT Lab % | Func. ICT Lab % |
|---|---|---|---|---|---|---|
| Arunachal Pradesh | 3,229 | 33.6 (−29.9) | 42.0 (−15.9) | 25.7 (−4.9) | 12.9 (−12.7) | 11.0 (−10.2) |
| Assam | 55,283 | 87.2 (+23.7) | 66.7 (+8.8) | 20.8 (−9.8) | 41.1 (+15.5) | 35.0 (+13.8) |
| Manipur | 4,666 | 36.6 (−26.9) | 35.2 (−22.7) | 18.6 (−12.0) | 49.1 (+23.5) | 32.5 (+11.3) |

| | | | | | | |
|---|---|---|---|---|---|---|
| Meghalaya | 14,587 | 26.4 (−37.1) | 17.5 (−40.4) | 4.3 (−26.3) | 6.3 (−19.3) | 4.8 (−16.4) |
| Mizoram | 3,974 | 64.9 (+1.4) | 64.7 (+6.8) | 11.3 (−19.3) | 22.7 (−2.9) | 20.1 (−1.1) |
| Nagaland | 2,750 | 59.0 (−4.5) | 78.6 (+20.7) | 32.8 (+2.2) | 14.7 (−10.9) | 11.8 (−9.4) |
| Sikkim | 1,245 | 88.9 (+25.4) | 97.3 (+39.4) | 80.1 (+49.5) | 89.2 (+63.6) | 62.4 (+41.2) |
| Tripura | 4,943 | 41.7 (−21.8) | 61.0 (+3.1) | 21.6 (−9.0) | 43.1 (+17.5) | 42.5 (+21.3) |
| **India (National)** | 14,71,473 | 63.5 | 57.9 | 30.6 | 25.6 | 21.2 |

## 4. Policy Architecture: NEP 2020, Bhashini, and the IndiaAI Mission

### 4.1 National Education Policy 2020: The Language Mandate

NEP explicitly mandates mother tongue as the medium of instruction through at least Class 5, with a recommendation to extend this to Class 8 and beyond. NEP 2020 further commits to the development of technology-based learning material in tribal and indigenous languages, recognizing that educational quality cannot be achieved through instruction in a language children do not speak at home (Ministry of Education, 2020). For NEP 2020's language mandate to be realized through digital infrastructure, AI-powered learning tools, voice-based educational assistants, or online content on the DIKSHA platform, the languages of instruction must first be available in machine-readable, digitized form. NEP 2020 commits to the goal; the Bhashini platform and the IndiaAI Mission are, in theory, the delivery mechanisms; the gap between the two is precisely the low-resource-language problem documented in this paper.

### 4.2 Bhashini and Adi-Vaani: Achievement and Gap

Bhashini's achievement in scaling AI-driven translation and speech recognition for all 22 scheduled Indian languages represents a genuine technological and institutional milestone. The platform's architecture, combining machine translation, text-to-speech, automatic speech recognition, and natural language understanding, provides a replicable model for language AI at scale (PIB, 2025). The Adi-Vaani extension to tribal languages, launched in 2024, demonstrates that the Indian government recognizes the inadequacy of a purely scheduled-language approach (DD News, 2025).

The limitation is one of language selection. Adi-Vaani's initial coverage of Santali, Bhili, Mundari, and Gondi reflects the tribal-language priorities of central India, not those of the Northeast. This is not surprising given that the Northeast's tribal languages are linguistically distinct, lack standardized scripts in many cases, and have smaller absolute speaker populations than the large central Indian tribal languages. But it means that the communities facing the most severe digital exclusion, the Khasi, Garo, Naga, and Arunachal language communities, remain unserved by India's flagship tribal AI platform even after its 2024 launch.

## 5. Indigenous Languages, Climate Knowledge, and  Case for Language AI in NE India

### 5.1 Indigenous Ecological Knowledge as Climate Data

Northeast India is among the most climate-vulnerable regions in India. The Eastern Himalayas, which span Arunachal Pradesh, Sikkim, and parts of Assam, are projected to lose over one-third of their glaciers by 2100 even under 1.5°C warming scenarios, with profound consequences for water security, agriculture, and

downstream flood patterns (IPCC, 2022). Assam, already subject to the most severe annual flooding of any Indian state, faces increasing flood frequency and intensity under climate change projections (ORF, 2024).

Within these communities, indigenous ecological knowledge (IEK) accumulated over generations of direct interaction with local ecosystems constitutes a significant and scientifically valuable dataset. Chinlampianga (2011) documented 15 distinct bioindicators used by Mizo communities for weather prediction, including plant phenology, animal behavior, and astronomical observations. The IJRISS study on climate change and SDG 13 in Mizoram (2025) identifies this IEK as a potentially significant complement to formal climate monitoring, particularly in areas with sparse meteorological infrastructure. The UNDP framework for integrating indigenous knowledge into climate change planning explicitly recognizes IEK as a form of climate intelligence that formal adaptation planning has systematically undervalued.

The critical point for this paper is that this knowledge exists almost entirely in oral, low-resource languages. Mizo bioindicators are encoded in Mizo vocabulary, ecological categories, and traditional practices that lack direct translation into Hindi or English and have no digital representation in any AI training corpus. As the climate crisis deepens, these knowledge systems face a double threat: the physical disruption of the ecosystems they describe, and the digital exclusion that prevents them from being recorded, shared, or integrated into formal climate response systems.

### 5.2 AI as a Tool for IEK Preservation and Climate Resilience

NLP tools designed for low-resource languages can serve a dual function in this context: they can preserve IEK by creating digital records in the original language, and they can translate and integrate this knowledge into formal climate adaptation frameworks. This is not a hypothetical application. The Masakhane NLP project in sub-Saharan Africa demonstrated that community-led, participatory data collection can build language AI tools in 60+ low-resource languages within a five-year horizon (Nekoto et al., 2020). The methodology training community members to record, transcribe, and annotate speech data in their own languages is directly transferable to the NE India context.

Google's Flood Forecasting Initiative, which has been operational in Assam and Bihar since 2020, demonstrates that AI-powered climate early warning can be deployed in the NE Indian context with measurable impact (Mateo-Garcia et al., 2021). However, this system currently operates in English and Assamese. Its extension to Bodo, Mizo, Khasi, and Naga languages, which would require exactly the kind of NLP infrastructure this paper argues is absent, would dramatically expand its reach into the most vulnerable tribal communities.

NASA and USAID's SERVIR program, operating across the Hindu Kush Himalaya region, provides satellite-based geospatial climate data for Arunachal Pradesh, Sikkim, and adjacent areas (servir.net). ICIMOD, the International Center for Integrated Mountain Development based in Kathmandu, has documented the integration of indigenous knowledge into climate adaptation planning across the Himalayan region, including NE India. These global frameworks demonstrate the institutional appetite for IEK integration; the missing infrastructure is language AI that can bridge oral indigenous knowledge and formal digital systems.

**6. An Integrated Framework for AI-Enabled Development in Northeast India**

*6.1 The Three-Pillar Integration*

The preceding analysis demonstrates that language, education, and climate are not separable policy domains in the context of NE India. Language AI serves as the foundational infrastructure. Without NLP tools in Khasi, Mizo, Bodo, Naga, and other NE languages, educational AI cannot deliver mother-tongue instruction, climate AI cannot access indigenous ecological knowledge (as UNDP and ICIMOD recommend), and government digital services cannot reach tribal communities in their own languages.

Education is the delivery mechanism. Schools are the most reliable and widely distributed infrastructure in remote NE communities. UDISE+ data shows that even in Meghalaya, with its severe digital deficit, over 14,500 schools exist as physical sites for digital access, teacher training, and community engagement. AI-assisted, mother-tongue educational tools delivered through the DIKSHA platform and Samagra Shiksha infrastructure can simultaneously advance learning outcomes, build AI literacy, and build teacher capacity for digital climate education.

Climate resilience is the urgent application. The climate crisis is not a future scenario for NE India; it is the present reality of Assam's annual floods, Mizoram's shifting agricultural seasons, and Arunachal's glacial retreat. AI-enabled climate early warning, crop advisory, and IEK documentation systems that operate in community languages can save lives, protect livelihoods, and ensure that generations of ecological knowledge are not lost as the ecosystems they describe are transformed.

**7. Conclusion**

This paper presents a quantitative and policy analysis of the structural relationship among language recognition, digital educational infrastructure, and indigenous knowledge preservation in Northeast India. The central findings are clear. States where the dominant language is constitutionally scheduled have, on average, 19 percentage points higher school internet access than states where the dominant language is non-scheduled, a gap that manifests across every digital infrastructure metric examined in UDISE+ 2024-25. Meghalaya, with its Khasi-speaking majority and 86% tribal population, records the last digital infrastructure performance in the region across all metrics, while Sikkim, with its Nepali-speaking majority and Eighth Schedule recognition, exceeds national averages on all five. The WMT 2024 shared task precisely quantifies data scarcity. Khasi has 14 times less NLP training data than Assamese, despite having a comparable speaker base.

These form a coherent picture of a structural condition in which language policy, specifically the decision about which languages to include in or exclude from constitutional recognition, has cascading effects across digital infrastructure, educational outcomes, AI capability, and the preservation of indigenous ecological knowledge. The 220+ languages of Northeast India are not merely communication systems; they are archives of ecological, cultural, and scientific knowledge accumulated over centuries. Their exclusion from AI training data is simultaneously an injustice, an epistemic loss, and a missed opportunity to develop AI systems that are useful to the NE communities.

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

---

AI Disclosure

Mostly Human with AI assistance >50

| Parts of your research | Score (1–4) |
|---|---|
| Idea generation | 3 |
| Literature selection | 3 |
| Literature review | 3 |
| Generation of research questions | 3 |
| Generation of a hypothesis | 3 |
| Research design (methods, data analysis, sampling) | 2 |
| Data collection | 2 |
| Data analysis and interpretation | 2 |
| Writing | 3 |
| Other (please specify) | - |
| **Your average score** | 24 |