# OpenReview forum: "Language, Artificial Intelligence, and Digital Equity in North East India: A Quantitative and Policy Analysis of Low-Resource Language Exclusion, Educational Infrastructure, and Climate Knowledge System"
_NortheastGenAI/2026/Workshop — NortheastGenAI 2026 Workshop Submission_

### Official Review · ~Badal_Nyalang1 · 2026-05-23
**Most empirically grounded paper — Accept, Best Paper candidate**

**Rating:** 8
**Confidence:** 5

**Review:**

**Relevance: Strong**
Excellent T3 fit, with meaningful T1 and T2 dimensions. The three-pillar framing (language AI, education, climate) is well-suited to this workshop and the NE India grounding is consistent and specific throughout.

**Plausibility: Strong**
This is the most empirically grounded paper in the batch. The UDISE+ data is real, the 19-percentage-point gap finding is concrete and original, and the Khasi vs. Assamese data scarcity comparison from WMT 2024 is a genuinely useful quantification. The Mizo IEK case study from Chinlampianga (2011) is appropriately used as illustrative evidence, not overclaimed.

**Novelty: Good**
The scheduled vs. non-scheduled language correlation with school internet access is the standout contribution — that specific framing has not been quantified at sub-national level before, as the authors themselves claim. The three-pillar integration is less novel but is argued coherently rather than just asserted.

**Clarity: Good**
Well structured and readable. The AI disclosure is honest but the average score calculation appears wrong — 24 divided by 9 items is not a standard score. Minor issue but worth flagging.

**Verdict: Accept**
Strong paper, candidate for Best Paper. The most policy-relevant and empirically grounded submission overall. The UDISE+ finding alone makes it worth presenting.

*This review was generated with AI assistance and checked by the workshop chairs.*

---

### Decision · Program_Chairs · 2026-05-23

**Decision:**

Accept

**Comment:**

An empirically grounded submission in the batch. The UDISE+ data analysis is concrete and original, and the correlation between scheduled vs. non-scheduled language status and school internet access is a finding that has not been quantified at this sub-national level before. The three-pillar framing is coherent and the regional grounding is consistent throughout.

A minor issue: the AI disclosure score calculation appears incorrect. Authors should verify and correct this if possible before presentation.

Decision: Accept